



# The Zero Emission Commitment Model Intercomparison Project (ZECMIP) contribution to CMIP6: Quantifying committed climate changes following zero carbon emissions

Chris D. Jones[1], Thomas L. Frölicher[2,3], Charles Koven[4], Andrew H. MacDougall[5], H. Damon Matthews[6], Kirsten Zickfeld[7], Joeri Rogelj[8,9], Katarzyna B. Tokarska[10], Nathan Gillett[11], Tatiana Ilyina[12], Malte Meinshausen[13,14], Nadine Mengis[7], Roland Séférian[15], Michael Eby[16]

[1]Met Office Hadley Centre, Exeter, EX1 3PB, UK
[2]Climate and Environmental Physics, Physics Institute, University of Bern, Bern, 3012, Switzerland
[3]Oeschger Centre for Climate Change Research, University of Bern, Bern, 3012, Switzerland
[4]Climate and Ecosystem Sciences Division, Lawrence Berkeley National Lab, Berkeley, CA 94720, USA
[5]St. Francis Xavier University, Antigonish, B2G 2W5, Canada
[6]Concordia University, Montreal, Quebec, H3G 1M8, Canada
[7]Simon Fraser University, Burnaby, V5A 1S6, Canada
[8]International Institute for Applied Systems Analysis (IIASA), 2361 Laxenburg, Austria
[9] Grantham Institute for Climate Change and the Environment, Imperial College, London SW7 2AZ, UK
[10]School of Geosciences, University of Edinburgh, Edinburgh, EH9 3FF, UK
[11]Canadian Centre for Climate Modelling and Analysis, Environment and Climate Change Canada, Victoria, BC, V8W 2Y2, Canada.
[12]Max Planck Institute for Meteorology, Bundesstraße 53, 20146 Hamburg, Germany
[13]Climate & Energy College, School of Earth Sciences, The University of Melbourne, Parkville 3010, Victoria, Australia
[14]Potsdam Institute for Climate Impact Research (PIK), Telegraphenberg, 14412 Potsdam, Germany
[15]Centre National de Recherches Météorologiques (CNRM), Université de Toulouse, Météo-France, CNRS, Toulouse, France
[16] School of Earth and Ocean Sciences, University of Victoria, Victoria, BC, V8W 2Y2, Canada

*Correspondence to*: Chris Jones (chris.d.jones@metoffice.gov.uk)



**Abstract.**

The amount of additional future temperature change following a complete cessation of $CO_2$ emissions is a measure of the unrealized warming to which we are committed due to $CO_2$ already emitted to the atmosphere. This "Zero Emissions Commitment" (ZEC) is also an important quantity when estimating the remaining carbon budget – a limit on the total amount

of $CO_2$ emissions consistent with limiting global mean temperature at a particular level. In the recent IPCC Special Report on Global Warming of 1.5°C, the carbon budget framework used to calculate the remaining carbon budget for 1.5°C included the assumption that the ZEC due to $CO_2$ emissions is negligible and close to zero. Previous research has shown significant uncertainty even in the sign of the ZEC. To close this knowledge gap, we propose the Zero Emissions Commitment Model Intercomparison Project (ZECMIP), which will quantify the amount of unrealized temperature change that occurs after $CO_2$

emissions cease and investigate the geophysical drivers behind this climate response. Quantitative information on ZEC is a key gap in our knowledge, and one that will not be addressed by currently planned CMIP6 simulations, yet it is crucial for verifying whether carbon budgets need to be adjusted to account for any unrealized temperature change resulting from past $CO_2$ emissions. We request only one top priority simulation from comprehensive general circulation Earth System Models (ESMs) and Earth System Models of Intermediate Complexity (EMICs) – a branch from the 1% $CO_2$ run with $CO_2$ emissions

set to zero at the point of 1000 PgC of total $CO_2$ emissions in the simulation – with the possibility for additional simulations, if resources allow. ZECMIP is part of CMIP6, under joint sponsorship by C4MIP and CDRMIP, with associated experiment names to enable data submissions to Earth System Grid Federation. All data will be published and made freely available.



## 1. Introduction

The Zero Emissions Commitment (ZEC), or the amount of global mean temperature change that is still expected to occur after a complete cessation of $CO_2$ emissions, is a key component of estimating the remaining carbon budget to stay within global warming targets as well as an important metric to understand impacts and reversibility of climate change (Matthews and Solomon, 2013). Much effort is put into measuring and constraining the TCRE - the Transient Climate Response to cumulative $CO_2$ Emissions (Allen et al., 2009; Matthews et al., 2009; Zickfeld et al., 2009; Raupach et al., 2011; Gillett et al., 2013; Tachiiri et al., 2015; Goodwin et al., 2015; Steinacher and Joos, 2016; MacDougall, 2016; Ehlert et al., 2017; Millar and Friedlingstein, 2018). The TCRE describes the ratio between $CO_2$-induced warming and cumulative $CO_2$ emissions up to the same point in time, but it does not capture any delayed warming response to $CO_2$ emissions beyond the point that emissions reach zero. When using the TCRE to derive the carbon budget consistent with a specific temperature limit, the ZEC is often assumed to be negligible and close to zero (Matthews et al., 2017; Rogelj et al., 2011, 2018). Constraints on ZEC have not been systematically researched so far, although both TCRE and ZEC are required to relate carbon emissions to the eventual equilibrium warming (Rogelj et al., 2018).

It has been shown that continued $CO_2$ removal by natural sinks following cessation of emissions offsets the continued warming that would result from stabilised $CO_2$ concentration (Matthews and Caldeira, 2008; Solomon et al., 2009; Frölicher and Joos, 2010; Matthews and Weaver, 2010; Joos et al., 2013). This is partly due to the ocean uptake of both heat and carbon sharing some similar processes and timescales and it is therefore expected to lead to ZEC being small (Allen et al., 2018; Ehlert and Zickfeld, 2017; Gillett et al., 2011; Matthews and Zickfeld, 2012). This has been shown to be a general result across a range of models (Gillett et al., 2011; Lowe et al., 2009; Matthews and Zickfeld, 2012; Zickfeld et al., 2013). Most such literature focused on long timescales (up to and beyond a century). This led IPCC SR15 (Rogelj et al., 2018) to make the assumption for the estimation of carbon budgets that for timescales up to a century ZEC was uncertain, yet centred around zero. More detailed studies, however, have shown that ZEC can be (a) non-zero, possibly of either positive or negative sign that may change in time during the period following emissions ceasing (Frölicher et al., 2014; Frölicher and Paynter, 2015); and (b) it is both state and rate dependent - i.e. it varies depending on the amount of carbon emitted and taken up by the natural carbon sinks, and the $CO_2$ emissions pathway of its emissions prior to cessation (Ehlert and Zickfeld, 2017; Krasting et al., 2014; MacDougall, 2019).

When we consider stringent climate targets, such as limiting global mean warming to 1.5 or 2°C, and in light of approximately 1°C warming to date and potential future warming from non-$CO_2$ greenhouse gases, an uncertainty in ZEC of 0±0.1°C already leads to a substantial uncertainty in the remaining carbon budget. Given the current central estimate of the TCRE of 1.6°C per 1000 PgC (Collins et al., 2013), each 0.1°C of warming equates to approximately 60 PgC of $CO_2$ emissions, or approximately



6 years of current fossil fuel emission rates (Le Quéré et al., 2018). It has therefore emerged that quantitative information on ZEC is a key gap in our knowledge, and one that is not filled by currently planned CMIP6 simulations.

ZECMIP aims to fill this gap as efficiently as possible. Thereby, ZECMIP will support the assessment of remaining carbon
budgets based on the CMIP6 simulations and supersede the current practice of applying a single model estimate of ZEC or an estimate from a limited number of studies from the literature. Much more preferable is to coordinate parallel studies, with Earth System General Circulation Models (ESMs) and Earth System Models of Intermediate Complexity (EMICs), to measure both TCRE and ZEC in a common scenario. Hence, we proposed using the 1% per annum increase in $CO_2$ concentration experiment (1pctCO2) from the CMIP6 Diagnostic Evaluation and Characterisation of Klima (DECK) simulations (Eyring et
al., 2016) as a common baseline simulation for estimating both the TCRE and the ZEC.

As a late addition to CMIP6, ZECMIP has been designed to address this important question with only one high priority simulation – **A1: "a zero-emission experiment following 1000 PgC emissions,"** implemented as a branch off the 1pctCO2 simulation from the point at which 1000 PgC in diagnosed cumulative emissions is reached. Additional simulations of lower
priority are also suggested which will aid further analysis. Branching from this idealised simulation avoids complications of non-$CO_2$ forcing and land-use or nitrogen deposition impacts on the carbon cycle, and also makes the ZEC quantified consistent with the TCRE values also derived from this simulation.

This paper documents the ZECMIP simulations, with a focus on the details needed for ESMs and EMICs to contribute the top
priority simulation of a ZEC run from the point of 1000 PgC emissions following 1% per year growth in $CO_2$.

ZECMIP analysis will draw on carbon cycle feedbacks and process understanding from C4MIP (Coupled Climate Carbon Cycle Model Intercomparison Project; Jones et al., 2016) and aims to complement analysis on reversibility and $CO_2$ removal under CDRMIP (Carbon Dioxide Removal Model Intercomparison Project; Keller et al., 2018). Both C4MIP and CDRMIP
encourage participation in the ZECMIP top priority simulation. For simplicity the data request is a replica of that for the CMIP6 emission-driven historical simulation (esm-hist). No new variables have been added. For EMICs the request is to output the same model variables as from the 1% run which forms the basis of ZECMIP, with the one addition of also providing atmospheric $CO_2$ concentration. Data can be published via the Earth System Grid Federation (ESGF) (for ESMs contributing to CMIP6). An equivalent data repository will be available for EMICs and likely based at University of Victoria – details will
communicated during summer 2019 via C4MIP and CDRMIP websites.



## 2. Simulation Protocol

Due to time pressures and limit in computational resources on modelling groups ZECMIP has just one high priority simulation, with a lower priority second simulation suggested (See Table 1). Other lower priority simulations are also detailed and welcomed. For EMIC model groups there is an extended protocol with longer and additional experiments. We welcome ESM

groups to also perform these additional simulations, but this is not required. Given that the overall CMIP6 protocol (Eyring et al., 2016) has been years in development, it is not possible to initiate a new MIP, nor allocate new CMIP tier-1 simulations during 2019. Instead, ZECMIP simulations are being included under C4MIP and CDRMIP and included in CMIP as tier-2 and tier-3 simulations so that they do not become mandatory "entry card" requirements for C4MIP or CDRMIP. Hence, our top priority simulation, A1, is classed as CMIP tier-2 simulation; all others are classified as tier-3 simulations. However, Table 1

lists the simulations prioritised by ZECMIP to guide groups who have limited resources to perform the simulations. We hope as many groups as possible perform as many of the simulations as possible, and participating model groups will be offered co-authorship on the manuscript containing the analysis to be submitted this year (by December 2019).

### 2.1. Simulation set – A: Abrupt-zero emissions

All ZECMIP simulations are required to be in "emissions-driven mode". Experiments under set "A" require branching off from a simulation where $CO_2$ concentration follows a 1% per annum increase from pre-industrial levels. This presents model groups with a choice of how to initialise experiments A1 to A3. Some models may have the capability to switch from concentration-driven to emissions-driven configuration, but some models may not, or model groups may not have confidence that they can do so without a shock to the model system. In the case of the former, the concentration-driven DECK 1pctCO2

simulation can be used to initiate experiments A1 to A3. Otherwise, models should perform simulation A0 to generate initial conditions for A1 to A3.

We do not specify a precise definition of how to make this choice but suggest that when an emissions-driven control run is initiated from a concentration driven control run, any subsequent change in atmospheric $CO_2$, major carbon stores, or global

temperature should all be approximately within the expected inter-annual variability of the control run.

**A0. "esm-1pctCO2".** Run an emissions-driven version of 1pctCO2 to get to the branch-off point for A1 to A3. The requirement to run this is a model-by-model decision. The compatible emissions timeseries for this simulation should be calculated from the 1pctCO2 and used to branch esm-1pctCO2 from esm-piControl to replicate the 1% profile as closely as

possible up to the desired cumulative emission before setting emissions to zero from this point.



The compatible emission rate $E$ (PgC yr$^{-1}$) can be calculated from the 1pctCO2 concentration-driven simulation, as described in Jones et al. (2013): see their section 2b. In summary, changes in atmospheric $CO_2$ concentration ($C_A$) are balanced by anthropogenic emissions, $E$, and changes in the natural land and ocean carbon reservoirs ($C_L$ and $C_O$ respectively). Therefore, the compatible emissions can be calculated simply as:

$$E = \frac{d}{dt}(C_{Tot}) = \frac{d}{dt}(C_A) + \frac{d}{dt}(C_L + C_O)$$

Where units of all quantities are in PgC. Changes in atmospheric $CO_2$ can be converted from concentration (ppm) to mass (PgC) by a simple scaling of 2.12. Typically, the time derivative $d/dt$, is taken to imply changes per year – i.e. annual changes in the carbon stores are used in order to calculate annual emission, $E$. The calculation is done using global total amounts. A model decision is required on the spatial pattern of emissions – we suggest globally uniform at surface. Models that have run multiple ensemble members for the concentration-driven 1pctCO2 experiment should use ensemble-mean values of $C_L$ and $C_O$ from those runs to derive the emissions for forcing the esm-1pctCO2 simulation. This will minimize the effect of interannual variability of carbon sinks on the diagnosed compatible emissions.

ZECMIP simulation set A is based on $CO_2$-only, 1% run (either concentration driven DECK: "1pctCO2", or the above described A.0 "esm-1pctCO2"), with all the other external forcing held at pre-industrial conditions (i.e., non-$CO_2$ greenhouse gases, aerosols, volcanoes, land-use changes, solar irradiance). After following the $CO_2$ concentration up to the level described below, branch off with prognostic $CO_2$ (a.k.a. "Emissions driven") but with carbon emissions set to zero (E=0). Simulate the subsequent reduction in atmospheric $CO_2$ and change in climate for at least 100 years.

Branch off at given cumulative emissions of:

- **A1. "esm-1pct-brch-1000PgC".** 1000 PgC. ZECMIP top priority simulation. This corresponds to approximately 2°C $CO_2$-induced warming above pre-industrial (with the year 1850 here taken as proxy for pre-industrial). This is the top priority ZECMIP simulation. Figure 1 shows example results from two models.
- **A2. "esm-1pct-brch-750PgC".** 750 PgC. This is a simulation corresponding to approximately 1.5°C $CO_2$-induced warming above 1850. Optional.
- **A3. "esm-1pct-brch-2000PgC".** 2000 PgC). This simulation will give insights in ZEC for a possible higher $CO_2$-induced warming. Optional.





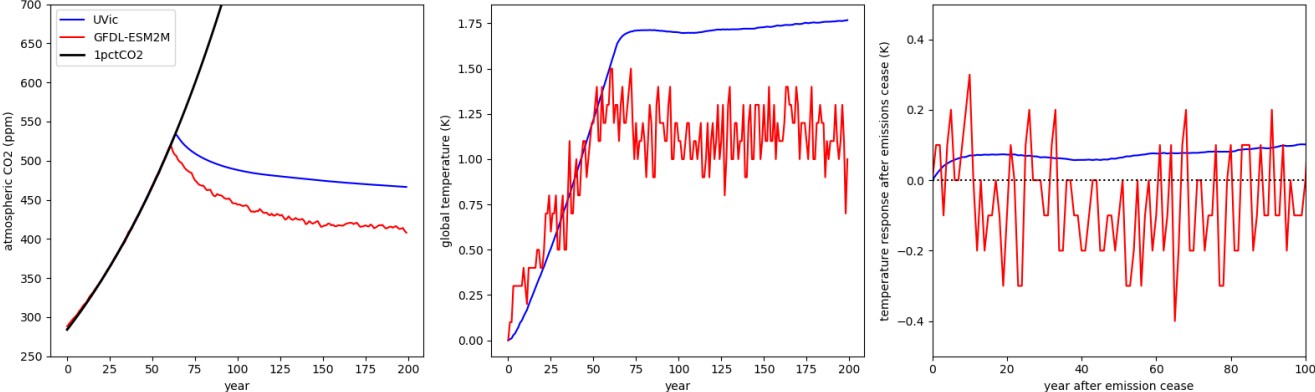

**Figure 1. Example results from simulation A1 from the UVIC ESCM (Weaver et al., 2001; MacDougall and Knutti, 2016; blue) and GFDL-ESM2M (Dunne et al., 2012, 2013; blue) models. (a) CO2 concentration prescribed (black line) in the 1pctCO2 simulation and simulated (red, blue lines) by the two models; (b) simulated global mean surface air temperature for the same period; (c) global mean temperature response from the branch point off the 1% simulation with zero subsequent emissions.**

The experimental design is for all models to branch off at a common cumulative carbon emission level, acknowledging that this will mean a different year for ceasing emissions and thus a slightly different atmospheric $CO_2$ concentration and departure of global mean temperature from 1850 for each model at the beginning of the ZECMIP simulations. EMICs should run the simulations for at least 1000 years. We anticipate that the small signal-to-noise ratio of the ZEC versus the internal climate variability may require ensemble of simulations. However, acknowledging ESM time pressure and limits in computational resources only one ensemble member is required.

Experiment A1 aims to quantify ZEC at 1000 PgC (cumulative emissions), at which point TCRE will be calculated. A2 and A3 explore the *state* dependence of ZEC at approximately 1.5°C $CO_2$-induced warming above 1850 and at significantly higher cumulative emissions respectively.

## 2.2. Simulation set – B: Bell-shape zero emissions

This second set of experiments, B1 to B3, aims to explore the dependence of ZEC on $CO_2$ emissions *rate* by following a pathway emitting the same cumulative emissions as A1 to A3 but with a smooth transition to zero emissions, followed by 100 years of E=0 (EMICs for at least 1000 years). The main purpose of this experiment is to quantify the dependency of ZEC on emission pathways and the emission rate prior to the point when TCRE is evaluated, as the Earth system is subject to comparatively low emissions, occurring just before the TCRE evaluation point of zero emission after 100 years of simulation – compared to the sudden cessation of high emissions in experiment A.1, A.2 and A.3.





These B-experiments are run in emissions-driven configuration ($CO_2$-only: following 1pctCO2 and piControl, all other external forcing is fixed at pre-industrial), assuming a "bell shaped" emissions profile (Figure 2). At end of 100 years emissions profile, simulations should continue with zero emissions for at least 100 years (for ESMs) and 1000 years (EMICs).

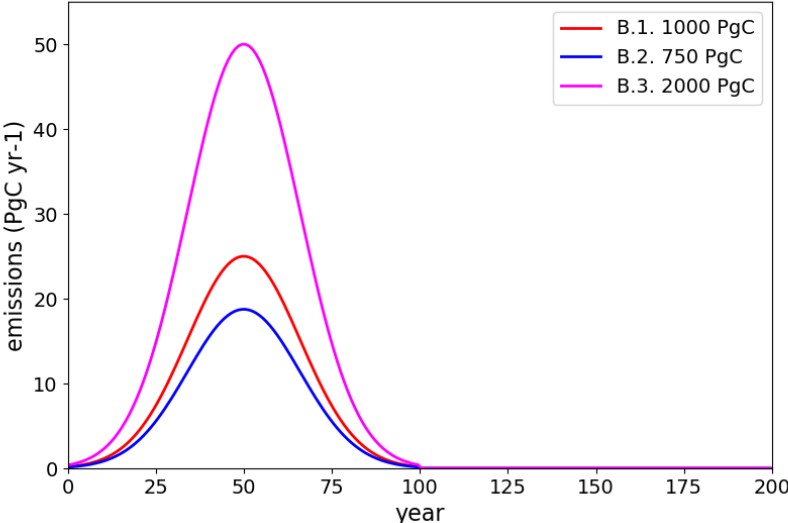

**Figure 2. Time series of global $CO_2$ emissions for bell curve pathways B1 to B3. The numbers in the legend indicate the cumulative amount of $CO_2$ emissions for each simulation.**

The bell-curve is designed to give cumulative emissions of:

- **B1. "esm-bell-1000PgC".** 1000 PgC

- **B2. "esm-bell-750PgC".** 750 PgC

- **B3. "esm-bell-2000PgC".** 2000 PgC

By design, this set of B-experiments utilise the same cumulative emissions as the respective simulations in set "A" experience up to their branch point. These emissions are applied over 100 years, followed by zero emissions for 100 years (ESMs) or

1000 years (EMICs). These additional simulations allow for a direct comparison of the two ZEC experiment sets, given the same amount of cumulative emissions. A model decision is required on the spatial pattern of emissions – we suggest globally uniform at surface. The timeseries of global $CO_2$ emissions for the above curves is listed in Appendix A and is hosted on the C4MIP (www.c4mip.net) and CDRMIP (https://www.kiel-earth-institute.de/CDR_Model_Intercomparison_Project.html) websites.



## 3. ZECMIP outlook and conclusions

The experiments outlined above will lay the foundation for coordinated multi-model analysis of the Zero Emissions Commitment. The absence of a dedicated experiment to quantify ZEC across CMIP models was identified and is addressed by our top priority experiment, A1. Investigations into the state, rate and pathway dependence of the ZEC are aided by further

experiments with sudden and gradual cessation of emissions. ZECMIP was motivated to keep the experiment design both lightweight and simple to follow, but in future, further simulations could be defined to explore additional issues such as cessation of emissions of non-$CO_2$ greenhouse gases, aerosols, or from land-use activities. The complexity of defining such experiments precluded an exhaustive inclusion in this first generation of ZECMIP but we acknowledge the importance of rate- and pathway dependency, as well non-$CO_2$ aspects in determining ZEC and the remaining carbon budget overall (MacDougall

et al., 2015; Rogelj et al., 2015; Mengis et al., 2018; Tokarska et al., 2018).

The requirement for specific information regarding ZEC to assess remaining carbon budgets was identified in the IPCC Special Report on Global Warming of 1.5°C (Rogelj et al., 2018). An initial paper exploring ZEC in this context, explicitly on timescales of relevance to 21[st] century carbon budgets, is planned on a timeline that could support an improved assessment of

the ZEC and its influence on carbon budgets in the IPCC Sixth Assessment. All participating model groups who are able to complete and provide data for simulation A1 in time will be invited to join this analysis.

ZECMIP welcomes community engagement in the participation of simulations and their analysis, and input to future analysis and experimental design. We hope to bring together ESMs and EMICs to enable analysis across timescales from decadal to

centennial to millennial.

Furthermore, as a set of numerical simulations, ZECMIP is intended to complement existing CMIP activity especially on carbon cycle feedbacks, $CO_2$ removal and reversibility of the climate system. C4MIP simulations aim to address model evaluation during the historical period from 1850 to present day, along with process-level feedback analysis. CDRMIP adds

to this with exploration of the processes controlling the response of the climate and carbon cycle to negative emissions, and reversibility of components of the Earth System. ZECMIP will contribute additional simulations and analysis to aid understanding of the mechanisms of the climate response to $CO_2$ emissions and relationships between transient and equilibrium climate sensitivities. We hope that ZECMIP analysis will address the crucial knowledge gap surrounding committed warming following ceasing emissions and provide valuable support for assessment of carbon budgets to achieve climate targets.


**Data availability**

As with all CMIP6-endorsed MIPs, the model output from the ZECMIP simulations described in this paper will be distributed through the Earth System Grid Federation (ESGF) with version control and digital object identifiers (DOIs) assigned. No additional model forcings are required beyond those already used for piControl and 1pctCO2 simulations apart from the emission inputs for the proposed B experiments which are described in Appendix A to this paper and are hosted on the C4MIP and CDRMIP websites.

**Author contributions**

All authors partook in workshop discussions to identify research needs and design the experimental protocol described here to address them. All authors contributed to the development of the manuscript.

**Competing interests.**

The authors declare that they have no conflict of interest.

**Acknowledgements**

This protocol was devised at a Global Carbon Project workshop supported by H2020 EU project CRESCENDO under grant agreement No 641816. CDJ was supported by the Joint UK BEIS/Defra Met Office Hadley Centre Climate Programme (GA01101). TLF acknowledges support from the Swiss National Science Foundation under grant PP00P2_170687. KZ and AHMD acknowledge support from the National Sciences and Engineering Research Council of Canada's Discovery Grant program. CDK acknowledges support from the US DOE BER Regional and Global Model Analysis Program through the Early Career Research Program and the RUBISCO SFA projects. K.B.T. was supported by the UK NERC-funded SMURPHs project (NE/N006143/1). We gratefully acknowledge Friedrich Burger for providing the GFDL-ESM2M data.



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



**Table 1.** ZECMIP simulations and priorities for ESMs and EMICs.

| ZECMIP experiment | CMIP6 experiment ID | Description | ESM priority (at least 100 years) | EMIC priority (1000 years) |
|---|---|---|---|---|
| A0 | *esm-1pctCO2* | An emissions-driven simulation (fully interactive $CO_2$), initiated from the esm-piControl using $CO_2$ emissions diagnosed from the 1pctCO2 experiment so that the emissions-driven run replicates as closely as possible the 1pctCO2 concentration profile. Required to create start conditions for A1-3. Not required if model can use DECK 1pctCO2. | If required | If required |
| A1 | *esm-1pct-brch-1000PgC* | A zero-emissions simulation (fully interactive $CO_2$), branched from the point in the 1pctCO2 experiment (or A0 above) when the cumulative carbon emissions reach 1000 PgC | 1 | 1 |
| A2 | *esm-1pct-brch-750PgC* | A zero-emissions simulation (fully interactive $CO_2$), branched from the point in the 1pctCO2 experiment (or A0 above) when the cumulative carbon emissions reach 750 PgC | 2 | 1 |
| A3 | *esm-1pct-brch-2000PgC* | A zero-emissions simulation (fully interactive $CO_2$), branched from the point in the 1pctCO2 experiment (or A0 above) when the cumulative carbon emissions reach 2000 PgC | | 2 |
| B1 | *esm-bell-1000PgC* | An emissions-driven simulation (fully interactive $CO_2$), initiated from esm-piControl using $CO_2$ emissions, amounting to 1000 PgC, following a bell-shape curve for 100 years followed by zero-emissions for at least 100 years | | 1 |
| B2 | *esm-bell-750PgC* | An emissions-driven simulation (fully interactive $CO_2$), initiated from esm-piControl using $CO_2$ emissions, amounting to 750 PgC, following a bell-shape curve for 100 years followed by zero-emissions for at least 100 years | | 2 |
| B3 | *esm-bell-2000PgC* | An emissions-driven simulation (fully interactive $CO_2$), initiated from esm-piControl using $CO_2$ emissions, amounting to 2000 PgC, following a bell-shape curve for 100 years followed by zero-emissions for at least 100 years | | 2 |

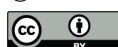



## Appendix A. CO$_2$ Emissions for Bell-curve simulations B1-3.

This table lists the global CO$_2$ emissions, in PgC yr$^{-1}$, to be applied for the first 100 years of simulations B1-3. This period should be followed by at least 100 years of zero emissions for ESMs and 1000 years for EMICs (see Figure 2). The spatial distribution of these emissions is not prescribed and is a free choice for model groups, but we suggest globally uniform at surface. This data in .csv format is available from the C4MIP (www.c4mip.net) and CDRMIP (https://www.kiel-earth-institute.de/CDR_Model_Intercomparison_Project.html) websites.

| year | B1. 1000 PgC | B2. 750 PgC | B3 2000 PgC |
|---|---|---|---|
| 1 | 0.20873014 | 0.1565476 | 0.41746028 |
| 2 | 0.25276203 | 0.18957153 | 0.50552407 |
| 3 | 0.30488921 | 0.22866691 | 0.60977842 |
| 4 | 0.3663328 | 0.2747496 | 0.73266561 |
| 5 | 0.43844296 | 0.32883222 | 0.87688592 |
| 6 | 0.52270172 | 0.39202629 | 1.04540343 |
| 7 | 0.62072365 | 0.46554273 | 1.24144729 |
| 8 | 0.73425378 | 0.55069034 | 1.46850756 |
| 9 | 0.86516239 | 0.64887179 | 1.73032477 |
| 10 | 1.01543611 | 0.76157709 | 2.03087223 |
| 11 | 1.18716509 | 0.89037382 | 2.37433018 |
| 12 | 1.38252556 | 1.03689417 | 2.76505111 |
| 13 | 1.6037577 | 1.20281828 | 3.2075154 |
| 14 | 1.8531385 | 1.38985388 | 3.706277 |
| 15 | 2.13294934 | 1.59971201 | 4.26589868 |
| 16 | 2.44543847 | 1.83407885 | 4.89087694 |
| 17 | 2.79277839 | 2.09458379 | 5.58555678 |
| 18 | 3.17701853 | 2.3827639 | 6.35403707 |
| 19 | 3.60003364 | 2.70002523 | 7.20006728 |
| 20 | 4.06346858 | 3.04760144 | 8.12693716 |
| 21 | 4.56868053 | 3.4265104 | 9.13736106 |
| 22 | 5.11667948 | 3.83750961 | 10.233359 |
| 23 | 5.70806844 | 4.28105133 | 11.4161369 |





| | | | |
|---|---|---|---|
| 24 | 6.34298476 | 4.75723857 | 12.6859695 |
| 25 | 7.0210441 | 5.26578308 | 14.0420882 |
| 26 | 7.74128883 | 5.80596662 | 15.4825777 |
| 27 | 8.50214249 | 6.37660687 | 17.004285 |
| 28 | 9.30137222 | 6.97602916 | 18.6027444 |
| 29 | 10.1360608 | 7.60204558 | 20.2721216 |
| 30 | 11.0025899 | 8.25194241 | 22.0051798 |
| 31 | 11.8966362 | 8.92247716 | 23.7932724 |
| 32 | 12.8131814 | 9.60988606 | 25.6263628 |
| 33 | 13.746537 | 10.3099028 | 27.493074 |
| 34 | 14.6903849 | 11.0177887 | 29.3807697 |
| 35 | 15.6378333 | 11.728375 | 31.2756666 |
| 36 | 16.5814888 | 12.4361166 | 33.1629776 |
| 37 | 17.5135425 | 13.1351569 | 35.027085 |
| 38 | 18.4258706 | 13.819403 | 36.8517412 |
| 39 | 19.3101466 | 14.48261 | 38.6202932 |
| 40 | 20.1579639 | 15.1184729 | 40.3159277 |
| 41 | 20.9609659 | 15.7207244 | 41.9219317 |
| 42 | 21.7109814 | 16.2832361 | 43.4219629 |
| 43 | 22.400162 | 16.8001215 | 44.8003239 |
| 44 | 23.0211173 | 17.265838 | 46.0422347 |
| 45 | 23.5670474 | 17.6752855 | 47.1340948 |
| 46 | 24.0318658 | 18.0238993 | 48.0637315 |
| 47 | 24.4103126 | 18.3077344 | 48.8206251 |
| 48 | 24.6980536 | 18.5235402 | 49.3961072 |
| 49 | 24.8917628 | 18.6688221 | 49.7835257 |
| 50 | 24.9891865 | 18.7418898 | 49.9783729 |
| 51 | 24.9891865 | 18.7418898 | 49.9783729 |
| 52 | 24.8917628 | 18.6688221 | 49.7835257 |
| 53 | 24.6980536 | 18.5235402 | 49.3961072 |
| 54 | 24.4103126 | 18.3077344 | 48.8206251 |
| 55 | 24.0318658 | 18.0238993 | 48.0637315 |





| | | | |
|---|---|---|---|
| 56 | 23.5670474 | 17.6752855 | 47.1340948 |
| 57 | 23.0211173 | 17.265838 | 46.0422347 |
| 58 | 22.400162 | 16.8001215 | 44.8003239 |
| 59 | 21.7109814 | 16.2832361 | 43.4219629 |
| 60 | 20.9609659 | 15.7207244 | 41.9219317 |
| 61 | 20.1579639 | 15.1184729 | 40.3159277 |
| 62 | 19.3101466 | 14.48261 | 38.6202932 |
| 63 | 18.4258706 | 13.819403 | 36.8517412 |
| 64 | 17.5135425 | 13.1351569 | 35.027085 |
| 65 | 16.5814888 | 12.4361166 | 33.1629776 |
| 66 | 15.6378333 | 11.728375 | 31.2756666 |
| 67 | 14.6903849 | 11.0177887 | 29.3807697 |
| 68 | 13.746537 | 10.3099028 | 27.493074 |
| 69 | 12.8131814 | 9.60988606 | 25.6263628 |
| 70 | 11.8966362 | 8.92247716 | 23.7932724 |
| 71 | 11.0025899 | 8.25194241 | 22.0051798 |
| 72 | 10.1360608 | 7.60204558 | 20.2721216 |
| 73 | 9.30137222 | 6.97602916 | 18.6027444 |
| 74 | 8.50214249 | 6.37660687 | 17.004285 |
| 75 | 7.74128883 | 5.80596662 | 15.4825777 |
| 76 | 7.0210441 | 5.26578308 | 14.0420882 |
| 77 | 6.34298476 | 4.75723857 | 12.6859695 |
| 78 | 5.70806844 | 4.28105133 | 11.4161369 |
| 79 | 5.11667948 | 3.83750961 | 10.233359 |
| 80 | 4.56868053 | 3.4265104 | 9.13736106 |
| 81 | 4.06346858 | 3.04760144 | 8.12693716 |
| 82 | 3.60003364 | 2.70002523 | 7.20006728 |
| 83 | 3.17701853 | 2.3827639 | 6.35403707 |
| 84 | 2.79277839 | 2.09458379 | 5.58555678 |
| 85 | 2.44543847 | 1.83407885 | 4.89087694 |
| 86 | 2.13294934 | 1.59971201 | 4.26589868 |
| 87 | 1.8531385 | 1.38985388 | 3.706277 |





| | | | |
|---|---|---|---|
| 88 | 1.6037577 | 1.20281828 | 3.2075154 |
| 89 | 1.38252556 | 1.03689417 | 2.76505111 |
| 90 | 1.18716509 | 0.89037382 | 2.37433018 |
| 91 | 1.01543611 | 0.76157709 | 2.03087223 |
| 92 | 0.86516239 | 0.64887179 | 1.73032477 |
| 93 | 0.73425378 | 0.55069034 | 1.46850756 |
| 94 | 0.62072365 | 0.46554273 | 1.24144729 |
| 95 | 0.52270172 | 0.39202629 | 1.04540343 |
| 96 | 0.43844296 | 0.32883222 | 0.87688592 |
| 97 | 0.3663328 | 0.2747496 | 0.73266561 |
| 98 | 0.30488921 | 0.22866691 | 0.60977842 |
| 99 | 0.25276203 | 0.18957153 | 0.50552407 |
| 100 | 0.20873014 | 0.1565476 | 0.41746028 |