# Peer review of "The Zero Emission Commitment Model Intercomparison Project (ZECMIP) contribution to C4MIP: Quantifying committed climate changes following zero carbon emissions"

_Geoscientific Model Development, 2019_

## Referee Comment (RC1) · Anonymous Referee #1 · 26 Jul 2019

[General comments] Jones et al. describe in this paper new experimental protocols for multi-model comparison study on Zero-emission commitment (ZEC) – global climate changes after future stoppage of anthropogenic CO2 emission. The authors design the protocols for Earth system models (ESM) and ESM of intermediate complexity (EMICs), to contribute to ongoing project "Coupled model intercomparison project phase 6 (CMIP6)". Because of urgent necessity in this science region and resource limitation of modeling centers, they propose a minimal set of experiments for evaluating ZEC in models.

As addressed by the authors, ZEC evaluation in models is an important and urgent issue for discussing remaining carbon budget for achieving specific mitigation goals. The scenario design for tier1 experiment is very simple – branching-off from 1%CO2 experiment by giving zero-emission, with free-evolving atmospheric CO2 concentration. This simplicity will be appreciated by many modeling centers, and the idealized scenario simulations are helpful when exploring underlying mechanism of ZEC. In addition, these protocol and simulation results will enable us to interpret ZEC in the context of transient climate response to cumulative emission (TCRE), which has been facilitated to approximate remaining carbon budget.

This paper is clearly written, and authors well summarizes the scientific question, experimental protocols and procedure in ZECMIP. Other comments are listed below, and all of them don't require much effort.

[Other Comments]

-P4, L3: Spell-out "CMIP6"

-P5, L26: about A0 experiment A0 experiment ("esm-1pctCO2") is an optional experiment, depending on the choice of modeling centers. Since A0 experiment seems not to be "tierized", I concern about the fate of the simulation output: do you expect modeling centers to submit A0 output to ESGF? Or do you have other plans for data archiving and sharing?

-P6, L9-12: about diagnosed compatible emission I propose another option to make diagnosed compatible emission without interannual variability – curve fitting to cumulative carbon emission, like,

1. Diagnose cumulative, not annual, carbon emission (CE)

2. Fit a curve to time–CE plots (like CE(t) = a*t + b*t^2 + c*t^3 + d*t^4)

3. Make annual emission from the fit curve

This method assures cumulative emission (if fitting is successful) and does not require multiple ensemble members.

-P7, Fig. 1 Label (a), (b), and (c) on panels

-P7, L19∼: Why do we need "bell-shaped" emission (smooth transition of emission rate) for discussing ZEC dependency on emission rate? Readers would be happy to see the rationale.

---

## Referee Comment (RC2) · Anonymous Referee #2 · 20 Aug 2019

Jones et al. describes a new, fast-track experiment, ZECMIP, under joint sponsorship of C4MIP and CDRMIP within CMIP6. The experiment is timely and of high relevance to on-going scientific discussions regarding methodological approaches for refining the definition of carbon budgets to meet certain policy-relevant global mean temperature goals. The authors propose a simplistic, but methodologically sound, approach to provide a scientific basis for understanding the effect of future warming or cooling after complete cessation of $CO_2$ emissions: the so-called "Zero Emissions Commitment" (ZEC).

[Figure]

This paper is novel, timely, and of high relevance to the audience of GMD. A multi-model comparison is of high importance due to the current lack of scientific consensus (order of magnitude, but also sign of effect).

Relevant comments are provided below, which mostly consist of clarification of expected results and possible pitfalls that could be observed during the experimental exercise.

1. P5L16-17. The authors note that "branching off" either from concentration-driven runs (experiment 1pctCO2) or emissions-driven runs (experiment A0) presents modelers with a decision or choice. What is the effect of choosing one option or the other? Do models that choose 1pctCO2 over A0 introduce additional uncertainty in the resulting estimation of ZEC?

2. P6L8-9. Another model decision described is the use of spatial patterns of emissions, where the authors suggest using a globally uniform pattern at the Earth's surface. What other options are available to the modelers? What effect on experimental results are expected from each choice? Is there a rational for choosing an option *other* than uniform-at-surface? These questions may be less relevant for experienced ESM developers, but would be enlightening for readers from other disciplines.

2a. Could a standard spatial pattern be derived from other CMIP6 MIPs (e.g., ScenarioMIP)? Would this provide any better consistency in ensemble comparison?

3. P8L1-2. Is it possible to be more precise as to the proposed distribution (Normal, Cauchy, Logistic, etc.) and associated parameters defining them?

3a. Similar to reviewer 1, please provide the rationale for choosing such a distribution. For example, why would a truncated log-normal distribution not be more appropriate (more emissions early with a decreasing tail profile)?

4. While the A-set of experiments seeks to provide a scientific basis for initial estimations of ZEC, the B-set of experiments also provides very important information and

context (i.e., do emissions rates significantly affect the estimation of ZEC). I would thus argue that it is also of high priority. While indeed there are resource and time limitations, it would be useful to the reader to understand what implications the lack of this information has on the estimation of ZEC. Is it possible to show results for the B1 experiment similarly to Figure 1? This would at least provide such context.

---

## Author Comment (AC1) · 6 Sep 2019

[General comments] Jones et al. describe in this paper new experimental protocols for multi-model comparison study on Zero-emission commitment (ZEC) – global climate changes after future stoppage of anthropogenic CO2 emission. The authors design the protocols for Earth system models (ESM) and ESM of intermediate complexity (EMICs), to contribute to ongoing project "Coupled model intercomparison project phase 6 (CMIP6)". Because of urgent necessity in this science region and resource limitation of modeling centers, they propose a minimal set of experiments for evaluating ZEC in models.

We thank the reviewer for their support of this important activity and recognising that we have kept requirements on model centres to a minimum which we believe will maximise participation in ZECMIP.

As addressed by the authors, ZEC evaluation in models is an important and urgent issue for discussing remaining carbon budget for achieving specific mitigation goals. The scenario design for tier1 experiment is very simple – branching-off from 1%CO2 experiment by giving zero-emission, with free-evolving atmospheric CO2 concentration. This simplicity will be appreciated by many modeling centers, and the idealized scenario simulations are helpful when exploring underlying mechanism of ZEC. In addition, these protocol and simulation results will enable us to interpret ZEC in the context of transient climate response to cumulative emission (TCRE), which has been facilitated to approximate remaining carbon budget.

This paper is clearly written, and authors well summarizes the scientific question, experimental protocols and procedure in ZECMIP. Other comments are listed below, and all of them don't require much effort.

Thank you.

[Other Comments]

-P4, L3: Spell-out "CMIP6"

Yes, we will do this on first usage

-P5, L26: about A0 experiment A0 experiment ("esm-1pctCO2") is an optional experiment, depending on the choice of modeling centers. Since A0 experiment seems not to be "tierized", I concern about the fate of the simulation output: do you expect modelling centers to submit A0 output to ESGF? Or do you have other plans for data archiving and sharing?

This is a good question. The A0 simulation is implicitly tierized as follows – if you don't need to do it then it's redundant (not tiered at all), but if you do need to do it then it is essential because A1 can't exist without it. This unfortunately doesn't match with giving it a tier number, but we will make clear in the text that A0 is considered a tier-1 experiment if it is required to achieve A1. Regarding the data submission – thank you for spotting this point. We will make it clear that data is required to be submitted for whichever run initialises A1. So if A0 is performed, then yes submission is required. We will clarify the text accordingly:

"We note that if simulation A0 is required to initialise the A1 simulation then it should be treated as equal priority to A1 and data submission to the ESGF is required."

-P6, L9-12: about diagnosed compatible emission I propose another option to make diagnosed compatible emission without interannual variability – curve fitting to cumulative carbon emission, like,

1. Diagnose cumulative, not annual, carbon emission (CE)

2. Fit a curve to time–CE plots (like CE(t) = a*t + b*t^2 + c*t^3 + d*t^4)

3. Make annual emission from the fit curve

This method assures cumulative emission (if fitting is successful) and does not require multiple ensemble members.

Thank you for this interesting suggestion. Although there is always tension between offering groups a choice (which may lead to inconsistency) or specifying a precise approach (which may in this case have noisy emissions), it is a good idea that groups may want to smooth their data. Some groups will want to use their "raw" emissions, or may have already done the runs, so we will keep this as an option. Rather than to specifically adopt these equations, our approach will be to mention that groups may choose to smooth their inferred emissions as long as the cumulative total agrees with 1000 PgC (or relevant branch points). We will modify the text accordingly:

"If desired, numerical smoothing of the global mean timeseries of emissions may also be applied as long as the cumulative total is not affected."

-P7, Fig. 1 Label (a), (b), and (c) on panels

Thank you  - we will add labels.

-P7, L19_: Why do we need "bell-shaped" emission (smooth transition of emission rate) for discussing ZEC dependency on emission rate? Readers would be happy to see the rationale.

It is an arbitrary choice which was easy to calculate. The key feature is a smooth transition to zero emissions in order to contrast with a sudden cessation. Similar Gaussian profiles were used by MacDougall and Knutti (2016, GRL). Given that we already show the numbers at an annual basis in the paper and hosted on the C4MIP website, readers do not need to make any calculation themselves. We will mention this arbitrary choice in the text, and in the Appendix provide the equation used to generate the profile:

"The data was calculated from a Gaussian curve according to:

$$E = k * \frac{1}{\sqrt{2\pi\sigma^2}} e^{-\frac{(x-\mu)^2}{2\sigma^2}}$$

Where emissions, E, are scaled by a constant, k, in order that the cumulative total matches the required amount for each scenario (1000 PgC for B1, 750 PgC for B2, 2000 PgC for B3). The parameters were set as μ=50 as the centre of the 100 year period, and σ=100/6 so that the distribution spans 3 standard deviations about the centre."

Jones et al. describes a new, fast-track experiment, ZECMIP, under joint sponsorship of C4MIP and CDRMIP within CMIP6. The experiment is timely and of high relevance to on-going scientific discussions regarding methodological approaches for refining the definition of carbon budgets to meet certain policy-relevant global mean temperature goals. The authors propose a simplistic, but methodologically sound, approach to provide a scientific basis for understanding the effect of future warming or cooling after complete cessation of CO2 emissions: the so-called "Zero Emissions Commitment" (ZEC).

This paper is novel, timely, and of high relevance to the audience of GMD. A multimodel comparison is of high importance due to the current lack of scientific consensus (order of magnitude, but also sign of effect).

We thank the reviewer for their support of this important activity and recognising the scientific novelty which we believe will maximise participation in ZECMIP.

Relevant comments are provided below, which mostly consist of clarification of expected results and possible pitfalls that could be observed during the experimental exercise.

Thank you for highlighting these issues which we hope we have clarified and helped model groups avoid pitfalls.

1. P5L16-17. The authors note that "branching off" either from concentration-driven runs (experiment 1pctCO2) or emissions-driven runs (experiment A0) presents modelers with a decision or choice. What is the effect of choosing one option or the other? Do models that choose 1pctCO2 over A0 introduce additional uncertainty in the resulting estimation of ZEC?

This is a fair question and we do not know for sure and it will likely vary from model to model. Our recommendation is to use the transition from concentration-driven to emissions-driven control runs as a guide, and then each group will be able to judge the required application of A0 or not. (Unpublished) Evidence from the UK model (UKESM1) is that differences in the 1% simulation from a small number of initial condition ensemble members is bigger that the change in $CO_2$ when we transition from concentration to emissions driven control runs. This corroborates our expectation that the choice of transition in ZECMIP simulations is not adding additional uncertainty to the results.

2. P6L8-9. Another model decision described is the use of spatial patterns of emissions, where the authors suggest using a globally uniform pattern at the Earth's surface. What other options are available to the modelers? What effect on experimental results are expected from each choice? Is there a rational for choosing an option *other* than uniform-at-surface? These questions may be less relevant for experienced ESM developers, but would be enlightening for readers from other disciplines.

2a. Could a standard spatial pattern be derived from other CMIP6 MIPs (e.g., ScenarioMIP)? Would this provide any better consistency in ensemble comparison?

This is a good point – there is no reason at all to deviate from this suggestion and in fact doing so may just confuse the analysis. In fact rather than suggesting alternative methods (such as from ScenarioMIP) we will go the other way and strengthen our "suggestion" of uniform emissions to a specification that this is what should be done:

Text revised to: "Emissions should be prescribed as globally uniform at the surface"

3. P8L1-2. Is it possible to be more precise as to the proposed distribution (Normal, Cauchy, Logistic, etc.) and associated parameters defining them?

3a. Similar to reviewer 1, please provide the rationale for choosing such a distribution. For example, why would a truncated log-normal distribution not be more appropriate (more emissions early with a decreasing tail profile)?

The key feature is to transition smoothly to zero emissions in contrast to the A1 sudden cessation. The rest of the profile is not crucial, so, in this stylised simulation we prefer to keep a simple symmetric profile. Similar Gaussian profiles were used by MacDougall and Knutti (2016, GRL). But we agree more description and justification is appropriate of this arbitrary choice. We will describe the distribution in the Appendix where we list the numbers – it is Gaussian, and we will provide the equation for clarity (see above response to reviewer#1).

4. While the A-set of experiments seeks to provide a scientific basis for initial estimations of ZEC, the B-set of experiments also provides very important information and context (i.e., do emissions rates significantly affect the estimation of ZEC). I would thus argue that it is also of high priority. While indeed there are resource and time limitations, it would be useful to the reader to understand what implications the lack of this information has on the estimation of ZEC. Is it possible to show results for the B1 experiment similarly to Figure 1? This would at least provide such context.

Thank you. We fully agree that it is also high priority. It would be great if all model groups could do this, and we will more strongly recommend in the text how useful it will be. In fact, if we had initiated this MIP earlier in the CMIP6 process we would very likely have also made this a tier-1 experiment. However, we are very mindful of the huge task faced by model groups to perform simulations and publish data against very challenging deadlines for the IPCC AR6 process and prefer to keep to our very minimal request of just one top priority simulation. We hope this will maximise participation in ZECMIP. We note that CMIP as an activity, and ZECMIP within it, will persist longer than IPCC AR6 and of course groups will have plenty of time to perform all the simulations and we can analyse them in the longer run. But it is crucial to get as many as possible to perform A1 in time for IPCC publication deadlines.

We do however fully agree we should provide more justification for the need for this simulation and we will expand the text to describe the implications of lacking it. We will add the following paragraph into section 2:

"The conventional way of estimating TCRE is using 1% CO2 model simulations. The tier-1 A1 simulation thus provides the most complementary and internally consistent quantification of the ZEC which is why we consider this to be the top priority. However, additional ZECMIP experiments with more gradually phased out emissions enable us to determine how the ZEC is expected to materialize over the timescales of more societally relevant CO2 emissions reduction rates. Analysis of pairs of "A" and "B" experiments will allow us to generalize the findings for other emission reduction pathways, allowing us to answer the question if temperature will continue to increase following a more realistic cessation of CO2 emissions."

**Other revisions in addition to Reviewer comments**

- To allow plotting of figure 1 and creation of the new figure 3, simulations with the GFDL-ESM2M model were required, which were performed by Friedrich Burger, and hence we have added his name to the author list

- By request of the CMIP panel we have made it clear in the title that ZECMIP contributes to CMIP via C4MIP rather than as a stand-alone MIP. It is too late in the CMIP6 process to endorse a stand-alone new MIP. Therefore minor change to title as:
"The Zero Emission Commitment Model Intercomparison Project (ZECMIP) contribution to C4MIP: Quantifying committed climate changes following zero carbon emissions"

- A few minor additions and edits to affiliations and acknowledgements